# Scalar Field Models of Barrow Holographic Dark Energy in $f(R,T)$ Gravity

Umesh Kumar Sharma [1,*], Mukesh Kumar [1] and Gunjan Varshney [2]

1    Department of Mathematics, Institute of Applied Sciences & Humanities, GLA University, Mathura 281 406, Uttar Pradesh, India
2    St. Francis Inter College, Hathras 204 101, Uttar Pradesh, India
*    Correspondence: sharma.umesh@gla.ac.in

**Abstract:** This research study investigates Barrow holographic dark energy with an energy density of $\rho_\Lambda = CH^{2-\Delta}$ by considering the Hubble horizon as the IR cut-off in the $f(R,T)$ gravity framework. We employ Barrow holographic dark energy to obtain the equation of the state for the Barrow holographic energy density in a flat FLRW Universe. Concretely, we study the correspondence between quintessence, $k$-essence, and dilation scalar field models with the Barrow holographic dark energy in a flat $f(R,T)$ Universe. Furthermore, we reconstruct the dynamics and potential for all these models for different values of the Barrow parameter: $\Delta$. Via this study, we can show that for Barrow holographic quintessence, $k$-essence, and dilation scalar field models, if the corresponding model parameters satisfy some limitations, the accelerated expansion can be achieved.

**Keywords:** BHDE; quintessence; $k$-essence; dilation; $f(R,T)$ gravity

## 1. Introduction

One of the most significant issues of modern cosmology is comprehending the behavior of dark energy (DE) [1–11]. The existence of a cosmological constant (CC) is the classical motivation for the current accelerated expansion of the universe [12,13]. However, an appealing choice is that DE is dynamic, occurring from an expansion of additional scalar field models. Because of the small value of suitable energy density, this opportunity becomes more realistic from the viewpoint of high-energy physics. Moreover, current conjectural conditions [14–16] firmly prefer cosmological models with more than one scalar field [17,18] for effective field hypotheses to be consistent with quantum gravity.

An association with the IR cut-off of a quantum field theory was established with the holographic principle (HP) beginning from the string hypothesis and black hole thermodynamics, associated with the energy of the vacuum and the maximum possible distance for this theory [19–22]. Specifically, to demonstrate the belatedly period of DE, the researchers use these holographic references vastly in the field of cosmology that is generally known as holographic dark energy (HDE) [23–29]. In the context of an early universe, the HP gained a significant importance that is parallel to the illustration of the DE period for both bounce and inflation scenarios [30–33]. The energy density of holography appears due to combining the HP with the dimensional analysis within the context of the development of holographic cosmology. Therefore, it turns the holographic cosmology differently and resembles the standard inflation or the models of DE (even if it is used for DE or inflation as well), which presented some appropriate higher curvature term(s) or scalar field term(s) in the Lagrangian.

Recently, J. D. Barrow presented Barrow entropy [34], which indicates that the fractal features might be presented via quantum gravitational effects on the structures of

the black hole, which will encode within the entropy process presented as $S \propto A^{1+\Delta}$ (A'' signifies the standard horizon area). The new exponent, $\Delta$, has the range $0 < \Delta < 1$ with $\Delta = 0$ corresponding to the standard smooth structure (in which case the Barrow entropy returns the standard Bekenstein–Hawking (BH) ones) and with $\Delta = 1$ corresponding to the most intricate structure. Hence, the application of this extended entropy relation as the basis of HDE gives rise to Barrow holographic dark energy (BHDE) [35,36], which is able to offer improved phenomenology compared to the standard scenarios of HDE. Lately, the Barrow entropy model of DE has gained plenty of attention to explain the DE epoch of the universe [37–58].

Illustrating the dynamical mechanism of the accelerated expansion of the cosmos, the transformation of general relativity (GR) is the most efficient strategy assuming the cosmological methods. Over the past decade, $f(R)$ theories (where the Hilbert–Einstein action is replaced with a more general function of the Ricci scalar) have been extensively studied as one of the simplest modifications relative to general relativity [59]. The researchers examined the astrophysical results to describe the cosmological mechanism of $f(R)$ gravity using different functional forms of $f(R)$ [60–64]. In the last decade, another modified theory of gravitation has been proposed by Harko et al. [65], called $f(R, T)$ gravity, where the gravitational Lagrangian is given by an arbitrary function of Ricci scalar $R$ and of the trace of the stress-energy tensor $T$. Note that the dependence from T may be induced by exotic imperfect fluids or quantum effects (conformal anomaly). The theory focuses on the Lagrangian matter because it includes a source term, which is a fundamental element of the field equations. This theory also recommends precise field equations for the issues of distinct matter research. Multiple models in this hypothesis [66–72] can correspond the different choices of functional forms in the $f(R, T)$ theory.

An interesting case where such a non-minimal interaction between curvature and matter is also admissible, having expected consequences for the covariant conservation law, is the family of the so-called $f(R, T)$ gravity [65]. There are, however, specific functional forms for $f(R, T)$ in which standard conservation can be preserved [67]. On the other hand, it is worth mentioning that, in this conservative subclass, there is no mixing involving the both dependencies on $R$ and $T$. In other words, Lagrangian density $f(R, T)$ admits the particular form $f(R, T) = f_1(R) + f_2(T)$. In this vein, it is possible to use the purely $T$-dependent term as part of a redefinition of the matter sector in a minimally coupled gravity [73,74]. This aspect helps us illustrate the close relation between the matter/curvature coupling and the conservation law to be obeyed by an energy–momentum tensor.

Viewing the scalar field DE models as an effective description of the underlying theory of DE and considering the holographic vacuum energy scenario as pointing in the same direction, it is interesting to study how the scalar field models can be used to describe the BHDE density as effective theories. Therefore, in this work, we reconstruct quintessence, $k$-essence, and dilation scalar fields connecting BHDE with these scalar field models. Considering the Barrow holographic density for the case of DE dominance, the explicit form of these scalar fields and potentials will be found and some comments about the cosmological dynamics will be given. Several researchers have worked with different choices for the scalar field models in GR and modified theories of gravity [75–88].

In this work, we are concerned with the reconstruction of quintessence, $k$-essence, and dilation scalar field models in the frame of $f(R, T)$ gravity that has not been researched before. The present work is mainly motivated by [80,81] and also inspired by [89]. The difference and similarity of the recent contribution with other works can be understood in a particular form [80,81,89]. Using the holographic method and examining the scalar field models of DE as an essential explanation of the underlying theory of DE, the authors of [89] proposed scalar field models to explain the holographic energy density by a new IR cut-off as $L^2 = \alpha H^2 + \beta \dot{H}$. They presented the correspondence between the $k$-essence, quintessence, dilation, and tachyon energy densities with the new holographic dark energy

(NHDE) in a flat FLRW universe in general relativity. Later, choosing the Hubble horizon as an IR cut-off, [90] explained the connection with the scalar field models, including dilation, tachyon, *k*-essence, and quintessence models. Assuming these scalar field models with the NHDE [80], the authors of [83] have generalized the work relative to the NHDE model in the Brans–Dicke theory. The paper studied the correspondence between the quintessence, chaplygin gas, dilation, tachyon, and *k*-essence scalar field models with NHDE in non-flat Brans–Dicke cosmology, and the dynamics and potentials have been reconstructed in this paper. Similar correspondences among DBI-essence, quintessence, and tachyon scalar field models and the characteristics of NHDE have been shown in the chameleon Brans–Dicke theory [81]. However, the given research shows some similarities and differences compared with other models [80,81,89]. Compared with standard general relativity or other theories of gravity, $f(R, T)$ gravity can show some differences in several problems of current interest [65].

The present paper is organized as follows. In Section 2, we review Barrow holographic dark energy. A brief review of the gravitational field equations of $f(R, T)$ gravity is provided in Section 3. In Section 4, the cosmological model and EoS parameter are presented. In Section 5, we examine the correspondence with scalar field models by dividing it into three subsections. The quintessence, *k*-essence, and dilation models for Barrow holographic dark energy are examined in Sections 5.1, 5.2, and 5.3, respectively. In Section 6, we provided a summary of our research.

## 2. Barrow Holographic Dark Energy

The main factor in the use of HP at the cosmological setup is that the universe horizon's (i.e., largest distance) entropy is proportional to its area, similarly to the BH entropy of a black hole. However, recently, Barrow showed that quantum-gravitational effects may introduce fractal and intricate features on the black-hole structure. This complex structure leads to finite volumes but has an infinite (or finite) area; this, therefore, results in a deformed black-hole entropy expression [34]:

$$S_B = \left(\frac{A}{A_0}\right)^{1+\Delta/2},$$ (1)

where $A$ denotes the standard horizon area, and $A_0$ denotes the Planck area. The deformation of quantum gravitation is quantified by new exponent $\Delta$, where $\Delta = 0$ corresponds to the standard BH entropy and $\Delta = 1$ is associated with the most complex and fractal structure. In contrast, inequality $\rho_\Lambda L^4 \leq S$ shows the HDE(standard), where $L$ represents the length of horizon and the application of the Barrow entropy Equation (1), which may found here by following $S \propto A \propto L^2$ [24], which lead us to

$$\rho_\Lambda = CL^{-(2-\Delta)},$$ (2)

where dimension $[L]^{-2-\Delta}$ is included with parameter $C$. For the case in which $\Delta = 0$, the standard HDE can be found by using equation $\rho_D = 3c^2 M_P^2 L^{-2}$, as shown by the equation mentioned above, where $C = 3c^2 M_p^2$, and $c^2$ is the model parameter ($M_p^2$ denotes the Planck mass). Moreover, in the case where the deformation effects quantified by $\Delta$ switch on, BHDE will depart from the standard one, leading to different cosmological aspects. If we consider the infrared cutoff ($L$) as Hubble horizon ($H^{-1}$), then the Barrow HDE density is obtained as follows.

$$\rho_\Lambda = CH^{2-\Delta}.$$ (3)

The quantum deformation and, hence, the deviation from BH entropy are quantified by Barrow exponent $\Delta$, which takes the value $\Delta = 1$ for maximal deformations, while $\Delta = 0$ is taken in the standard, non-deformed case. Anagnostopoulos et al. [43] used observational data from the direct measurements of the Hubble parameter from the cosmic chronometer

(CC) sample and observational data from the Supernovae (SNIa) Pantheon sample in order to extract constraints on the scenario of BHDE. They found that the standard value $\Delta = 0$ is inside the $1\sigma$ region; however, the mean value is $\Delta = 0.094$. They observed a value of $H_0 = 0.6895^{+0.0187}_{-0.0189}$ for the Hubble rate that is consistent with Planck data [91]. In this work, we used the value of model parameter $\Delta$ within the limit constrained in [43], and the value of the Hubble parameter from the Planck results [91].

### 3. Assessment of Gravitational Field Equations of $f(R, T)$ Theory

Let us consider the general reconstruction method for modified gravity with $f(R, T)$ theory action given as follows [65]

$$S = \frac{1}{16\pi} \int f(R, T) \sqrt{-g} \, d^4x + \int L_m \sqrt{-g} \, d^4x, \tag{4}$$

where $f(R, T)$ is an arbitrary function of Ricci scalar $R$ and the trace of the energy momentum tensor $T$. $L_m$ represents the Lagrangian matter density, and we define the stress–energy tensor of matter as follows [92].

$$T_{\mu\nu} = -\frac{2}{\sqrt{-g}} \frac{\delta(\sqrt{-g}L_m)}{\delta g^{\mu\nu}}, \tag{5}$$

The researchers provided three explicit formulations in the form of the functional relation of $f(R, T)$ as presented in [65].

$$f(R, T) = R + 2f(T), \quad f(R, T) = f_1(R) + f_2(T), \quad f(R, T) = f_1(R) + f_2(R)f_3(T) \ .$$

Different hypothetical models can be obtained for different choices of $f$ in the field equations of $f(R, T)$ gravity due to their dependence on the underlying physical nature of the matter field. Various researchers presented the cosmological consequences of the special class $f(R, T) = R + 2f(T)$ [93–99]. Taking the functional form $f(R, T) = R + 2f(T)$, the authors [100] aspired the cosmological result of a ghost DE model with sign-changeable interactions in $f(R, T)$ gravity. The main motivation to choose this special form of $f(R, T)$ gravity is that we obtain better results in studying correspondences between the quintessence, $k$-essence, and tachyon energy density with the Barrow holographic dark energy density in a flat FLRW Universe. Therefore, we considered the cosmological results of this class, i.e., $f(R, T) = R + 2f(T)$, in this work. The gravitational field equations immediately follow from Equation (4), and they are given by

$$R_{\mu\nu} - \frac{1}{2}Rg_{\mu\nu} = 8\pi T_{\mu\nu} + 2f'_T(T)T_{\mu\nu} + f(T)g_{\mu\nu}, \tag{6}$$

where prime denotes a derivative with respect to the argument. These field equations were suggested in [101] to solve the CC problem. The simplest cosmological model may be found by assuming a dust universe ($\rho = T$, $p = 0$) and by selecting a function $f(T)$ so that $f(T) = \lambda T$, where $\lambda$ is a constant.

### 4. The Cosmological Model

The flat FLRW metric can provide the metric of the universe as

$$ds^2 = -dt^2 + a^2(t)(dx^2 + dy^2 + dz^2) \tag{7}$$

The metric provided in Equation (7) provides the field equation of $f(R, T)$ gravity as

$$3\frac{\dot{a}^2}{a^2} = \rho_\Lambda(3\lambda + 8\pi), \tag{8}$$

$$2\frac{\ddot{a}}{a} + \frac{\dot{a}^2}{a^2} = \lambda\rho_\Lambda. \tag{9}$$

Hence, this $f(R, T)$ theory model is equivalent to a cosmological model with an effective cosmological constant $\Lambda_{eff} \propto H^2$, where $(H = \frac{\dot{a}}{a})$ is the Hubble function [101]. It is also interesting to note that generally for this choice of $f(R, T)$, the gravitational coupling becomes an effective and time-dependent coupling of the form $G_{eff} = G \pm 2f'(T)$. Thus, term $2f(T)$ in the gravitational action modifies the gravitational interaction between matter and curvature, replacing $G$ by a running gravitational coupling parameter.

The field equations reduce to a single equation for $H$ [65].

$$2\dot{H} = -3(\frac{8\pi + 2\lambda}{8\pi + 3\lambda})H^2, \tag{10}$$

We can construct the general solution as

$$H(t) = \frac{2}{3}\frac{(3\lambda + 8\pi)}{(2\lambda + 8\pi)}\frac{1}{t}. \tag{11}$$

The power-law expansion $a \propto t^\beta$ is presented by the scale factor and shows the correspondence with $t^\beta = a(t)$ (where $\beta = \frac{2(8\pi + 3\lambda)}{3(8\pi + 2\lambda)}$). While presenting the GO cut-off, for the small limit of $t$ in [26] with the future event horizon cut-off in [23], researchers obtained similar expressions for $H$ in [89]. On the other hand, the conservation equation is as follows.

$$\frac{\partial \rho_\Lambda}{\partial t} + 3H(\rho_\Lambda + p_\Lambda) = 0, \tag{12}$$

Applying the barotropic EoS and the density of pressure for HDE $p_\Lambda = \rho_\Lambda \omega_\Lambda$, we obtain an expression for the EoS parameter $\omega_\Lambda$ as

$$\omega_\Lambda = -1 + \frac{(\Delta - 2)}{3}\frac{\dot{H}}{H^2}. \tag{13}$$

Employing Equation (12) in Equation (13), we obtain

$$\omega_\Lambda = -1 + \frac{(2 - \Delta)}{2}\left[\frac{8\pi + 2\lambda}{8\pi + 3\lambda}\right], \tag{14}$$

The constraints of parameter $\Delta$ and $\lambda$ can be acquired by applying $\omega_\Lambda > -1$ as it is the permitted limit for quintessence [10], $k$-essence [102], and dilation [103] scalar field models. For the accelerated expansion of the universe, the EoS lies at $\omega_\Lambda > -1$ [10,102,103]. Equation (14) explains the EoS $\omega_\Lambda$ in terms of exponents $\Delta$ and $\lambda$. For the expanded acceleration of the universe, both constraints $\Delta$ and $\lambda$ must explain the constraints observed from Equation (14). We consider the $\omega_\Lambda > -1$ phase for $0 < \Delta < 1$ and $-6\pi < \lambda < -4\pi$ and a phantom with $\omega_\Lambda < -1$ for $\Delta > 2$ and $-6\pi < \lambda < -4\pi$, whereas EoS $\omega_\Lambda = -1$ imitates the cosmological constant for $\Delta = 2$ and $\lambda = -4\pi$.

Confinement is required by distinguishing the Barrow HDE from the standard HDE; the Barrow HDE delivers a more dynamical nature and quantifies the existence of non-extensive parameter $\Delta$. Due to its agreeable support, one can accept the sub-case of the framework of the standard HDE that is distinctive for $\Delta = 0$ [35]. Before the possibilities are assumed to be a promising contender, one should work on many types of research to explain the nature of DE. We ought to perform the study of the current universe in detail, seeing its evolution [35] and the underlying situations individually to categorize the structure of universal highlights in the present period. In this work, we assumed the model parameter value as $C = 3$ and $\Delta = 0.3, 0.5, 0.7, 0.9$, as limited in [38,39,43].

## 5. Correspondence with Scalar Field Models

In this section, we will study the correspondence between the quintessence, *k*-essence, and dilation scalar field models with BHDE in the framework of a flat $f(R, T)$ gravity. We will also reconstruct the potentials and the dynamics for these scalar-field models. We can give the related results of scalar fields and potentials for the BHDE model in the flat $f(R, T)$ gravity. In order to establish this correspondence, we compare the energy density of the BHDE model with the corresponding energy density of the scalar-field model, and we also equate the EoS for these scalar models with the EoS for the BHDE model.

### 5.1. Quintessence Model for Barrow Hologaphic Dark Energy

In the flat $f(R, T)$ gravity, we examine the correspondence between quintessence and BHDE in this section. The correspondence between BHDE and the quintessence scalar region is reconstructed by comparing the EoS parameter of the quintessence region with the BHDE model in (14). Moreover, the energy density of BHDE given in Equation (3) is equated with the respective energy density of the quintessence model. Similarly, for the quintessence model, the dynamics and potentials are reconstructed. We may form the relevant results for the potential and quintessence scalar region in the BHDE model in $f(R, T)$ gravity. For the quintessence field, the pressure and energy density can be read as [10] in the FLRW Universe.

$$p_q = -V_q(\phi) + \frac{1}{2}\dot{\phi}_q{}^2, \tag{15}$$

$$\rho_q = V_q(\phi) + \frac{1}{2}\dot{\phi}_q{}^2$$

The EoS parameter for the quintessence model of the scalar field is as follows

$$\omega_\phi = \frac{-2V_q(\phi) + \dot{\phi}_q{}^2}{2V_q(\phi) + \dot{\phi}_q{}^2}, \tag{16}$$

Comparing Equation (16) with the EoS parameter of BHDE given in Equation (14), we obtain

$$\frac{-2V_q(\phi) + \dot{\phi}_q{}^2}{2V_q(\phi) + \dot{\phi}_q{}^2} = -1 - \left(\frac{\Delta - 2}{2}\right)\left[\frac{8\pi + 2\lambda}{8\pi + 3\lambda}\right] \tag{17}$$

Equalizing both Equations (3) and (15), we obtain the following.

$$\rho_q = V_q(\phi) + \frac{1}{2}\dot{\phi}_q{}^2 = CH^{2-\Delta} \tag{18}$$

Upon solving the above equation, we obtain an expression for the scalar field potential as follows.

$$\phi_q = \pm\frac{2}{\Delta}\left[-C\left(\frac{\Delta - 2}{2}\right)\left(\frac{2}{3}\right)^{(2-\Delta)}\left(\frac{8\pi + 2\lambda}{8\pi + 3\lambda}\right)^{(3-\Delta)}\right]^{\frac{1}{2}} t^{\frac{\Delta}{2}} \tag{19}$$

Equation (19) gives an expression for the scalar field, and the graph is plotted in Figure 1 for both positive and negative scalar fields as (a) and (b), respectively. Figure 1a depicts that as redshift $z$ increases and scalar field $\phi_q$ decreases, and it portrays the constant at high redshifts and evolves finitely in the future. A similar nature was observed for scalar field $\phi$ in [104] for the *k*-essence scalar field [102]. For the negative scalar field, Figure 1b explains the shift in the scalar field. Due to this result, the potential's shape can be changed without having any impact on the expansion of the universe. By utilising Equation (11) in

Equation (18) and using this result with Equation (17), we obtain the expression in terms of scalar field $\phi$ for the potential field as follows.

$$V_q(\phi) = C\left[\left[\frac{2}{3}\left(\frac{8\pi+3\lambda}{8\pi+2\lambda}\right)\right]^{2-\Delta} + \left(\frac{\Delta-2}{4}\right)\left(\frac{2}{3}\right)^{2-\Delta}\left(\frac{8\pi+2\lambda}{8\pi+3\lambda}\right)^{3-\Delta}\right]A, \qquad (20)$$

$$A = \left[\left(\frac{2}{\Delta\phi_q}\right)^{\frac{2}{\Delta}}\left[-C\left(\frac{\Delta-2}{2}\right)\left(\frac{2}{3}\right)^{(2-\Delta)}\left(\frac{8\pi+2\lambda}{8\pi+3\lambda}\right)^{(3-\Delta)}\right]^{\frac{1}{\Delta}}\right]^{2-\Delta}$$

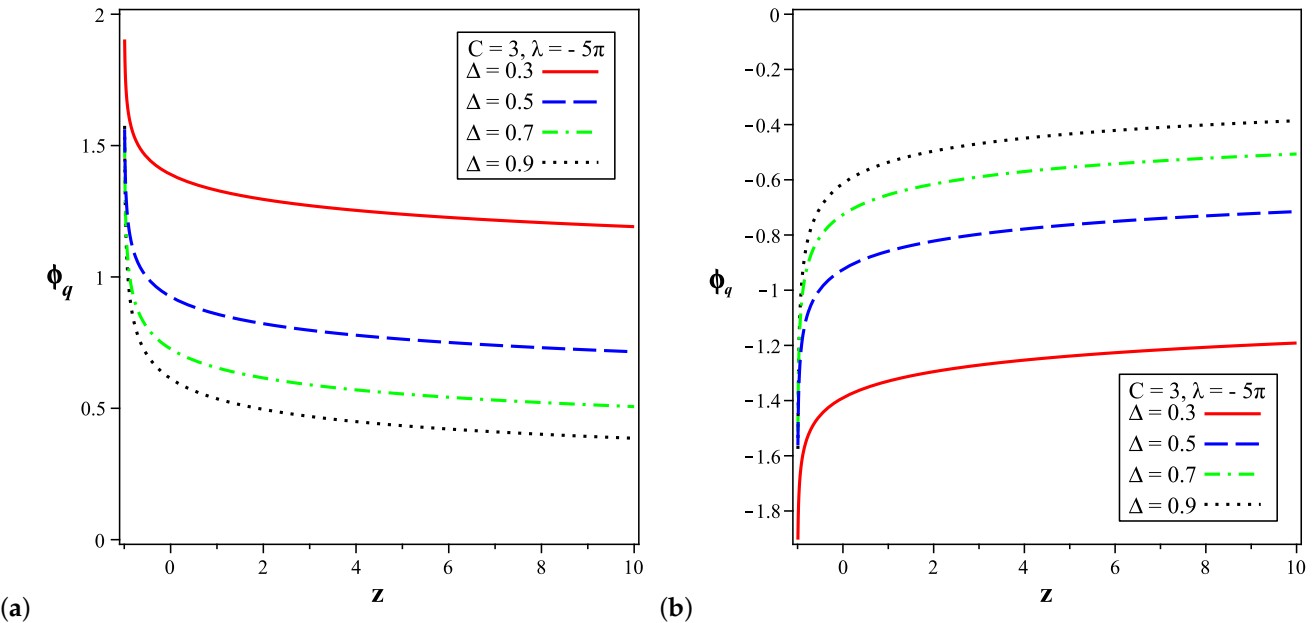

**Figure 1.** Depicting the nature between the redshift($z$) and the Barrow holographic quintessence scalar field ($\phi_q$), (**a**) ($\phi_q$) is positive and (**b**) ($\phi_q$) is negative; here, $C = 3$ and $\lambda = -5\pi$.

Figure 2 depicts the relation between quintessence potential $(V_q(\phi))$ and redshift $(z)$. It is clear from Figure 2 that from the past to the future, the potential for quintessence declines. The same nature was marked for the potential of HDE for the quintessence in [77] along with the tachyon model [76]. The quintessence potential $((V_q(\phi))$ with the scalar field $(\phi_q)$ is plotted in Figure 3. Figure 3a explains the expansion in the scalar field as the potential drops when the scalar field is positive. The same nature of the quintessence potential is demonstrated in [105], whereas when the scalar field is negative, Figure 3b then shows that in the past, the potential is steeper, and in the future, approaches to be flat, which means that the quintessence field is proceeding the potential down with the expansion of the Universe. [77] gave a similar result, and hence, the potential decreases along with the evolution of the universe, and it falls under the runway type, as suggested in [105].

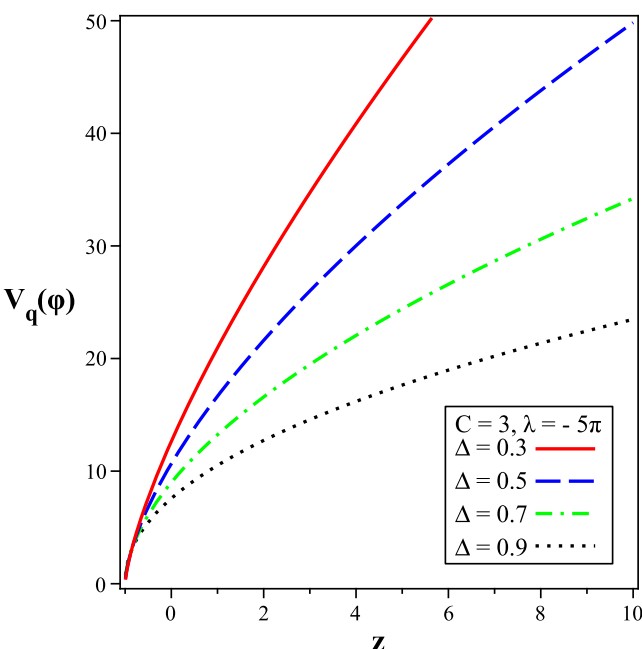

**Figure 2.** The plot of the Barrow holographic quintessence potential versus redshift ($z$); here, $C = 3$ and $\lambda = -5\pi$.

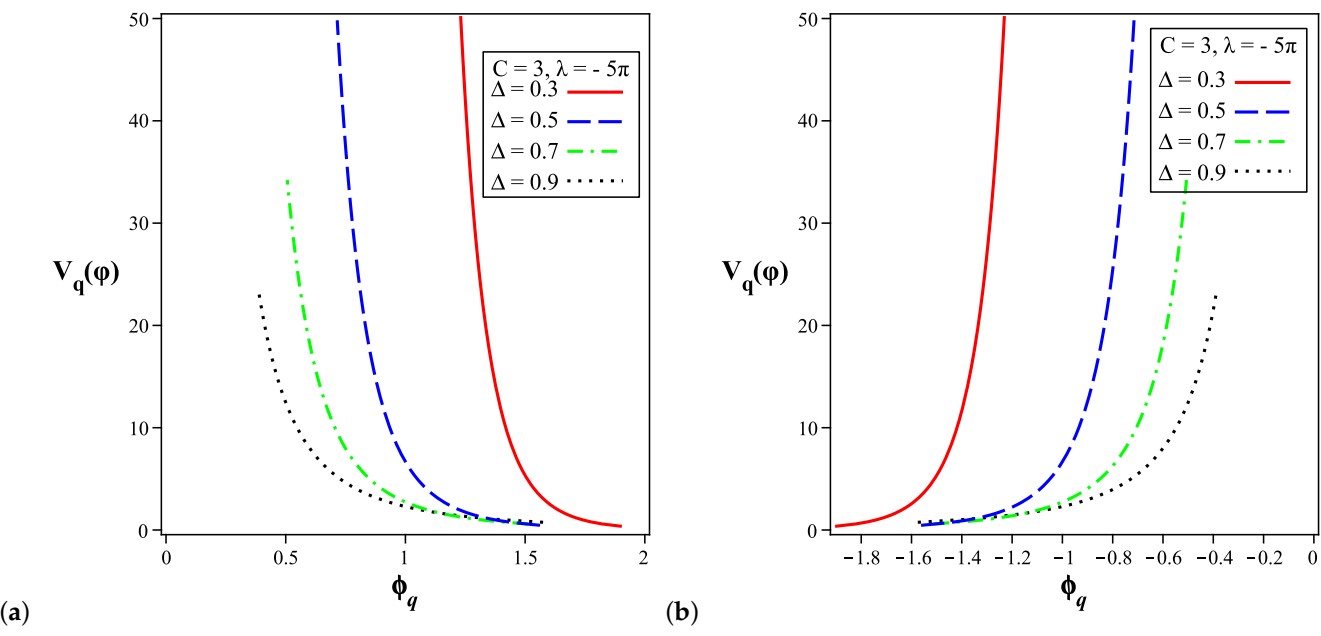

**Figure 3.** The behaviour of the Barrow holographic quintessence potential ($V_q(\phi)$) versus the scalar field ($\phi$): (**a**) ($\phi_q$) is positive and (**b**) ($\phi_q$) is negative; here, $C = 3$ and $\lambda = -5\pi$.

### 5.2. k-Essence Model for Barrow Holographic Dark Energy

This section presents the correspondence of the *k*-essence scalar field with BHDE. *k*-essence is one of the scalar field models used to explain the late-time acceleration of the universe. Concerning the scalar field [102], fine-tuning is required for the primary conditions because *k*-essence scenarios have attractor-like dynamics. The *k*-essence model is characterized by the action of the universal scalar field, a function $X = -\frac{1}{2\partial_\mu\phi\partial^\mu\phi}$, and $\phi$ is the non-standard kinetic terms described as [106]

$$S = \int d^4x \; p_k(\phi, X) \; \sqrt{-g}, \tag{21}$$

where $p_k(\phi, X)$ is the pressure density and generally restricts the Lagrangian density for the framework of $p_k(\phi, X) = g(X)f(\phi)$. We can change the Lagrangian density after studying the action of string theory on the low energies in [107] as

$$p_k(\phi, X) = (X^2 - X)f(\phi). \tag{22}$$

The Lagrangian density can create the energy–momentum tensor for the energy density of the scalar field [107].

$$\rho_k(\phi, X) = f(\phi)(-X + 3X^2). \tag{23}$$

The EoS parameter for the scalar field of *k*-essence can be obtained by equating Equation (22) with (23).

$$\omega_k = \frac{X - 1}{3X - 1}. \tag{24}$$

The cosmological constant EoS translates $X = 1/2$. $\omega_k < -1/3$ inclines toward the expanded acceleration, if $1/2 < X < 2/3$. We correspond the energy density of *k*-essence and the Barrow holographic model to the equivalent EoS parameter and the energy density of the Barrow holographic model to establish conformity with the EoS parameter. We make a comparison of the Barrow holographic EoS parameter using Equations (14) and (24) ($\omega_k = \omega_\Lambda$) to obtain $X$:

$$X = \frac{\lambda(4 + \Delta) + 4\pi(2 + \Delta)}{4\pi(2 + 3\Delta) + 3\lambda(2 + \Delta)}, \tag{25}$$

where $X$ is a constant. Certainly, in Equation (24), the EoS parameter varies for $X = 1/3$. The expanded acceleration inclines due to condition $2/3 > X$. We can solve Equation (25) to obtain the expression of the scalar field in a flat FLRW universe.

$$\phi = \pm\sqrt{2\left[\frac{\lambda(4 + \Delta) + 4\pi(2 + \Delta)}{4\pi(2 + 3\Delta) + 3\lambda(2 + \Delta)}\right]} \; t, \tag{26}$$

The integration constant is zero here. We plotted scalar field ($\phi$) versus redshift ($z$) in Figure 4. Figure 4a shows the plot when the scalar field is positive and (b) when the scalar field is negative. Figure 4a expresses the increment in redshift $z$ as the scalar field decreases, and at low redshift, it becomes finite, which was discussed in [104] with the GO cut-off for the holographic *k*-essence. In contrast, Figure 4b presents the expansion in the scalar field with a decrease in the redshift. Applying the resemblance among the *k*-essence and Barrow holographic energy density and Equations (3) and (23) ($\rho_\Lambda = \rho_k(\phi, X)$), and substituting $H$ by (11) and $X$ by Equation (25), we can acquire the potential, $f(\phi)$, for the *k*-essence Barrow holographic model as follows.

$$f(\phi) = C\left[\frac{2\sqrt{2}}{3\phi}\left(\frac{8\pi + 3\lambda}{8\pi + 2\lambda}\right)\right]^{2-\Delta}\left[\frac{\lambda(4 + \Delta) + 4\pi(2 + \Delta)}{4\pi(2 + 3\Delta) + 3\lambda(2 + \Delta)}\right]^{1-\Delta}\left[\frac{4\pi(2 + 3\Delta) + 3\lambda(2 + \Delta)}{6\lambda + 16\pi}\right], \tag{27}$$

Here, Equation (26) is used for $\phi$. Therefore, we acquire the *k* essence potential $f(\phi)$ by Equation (27) and the correspondence of the Barrow holographic *k*-essence. We can mark $\dot{\phi}$ = constant in Equation (26). This implies that the change in the *k*-essence kinetic term is

constant, but $\phi$ is affected with time and is not a constant. The graph detailing the potential versus redshift is plotted in Figure 5. We can note the decrease in the potential if we roll from the past to the future. Figure 6 shows the plot of thepotential against the scalar field. Here, Figure 6a presents the observation in which the scalar field is positive, and Figure 6b shows a negative scalar field. From Figure 6a, we can note that the potential drops with an expansion in the scalar field, whereas we can see the increment in the potential with the decrease in the scalar field in Figure 6b. A similar nature of the potential can be seen for Figures 5 and 6 in [104].

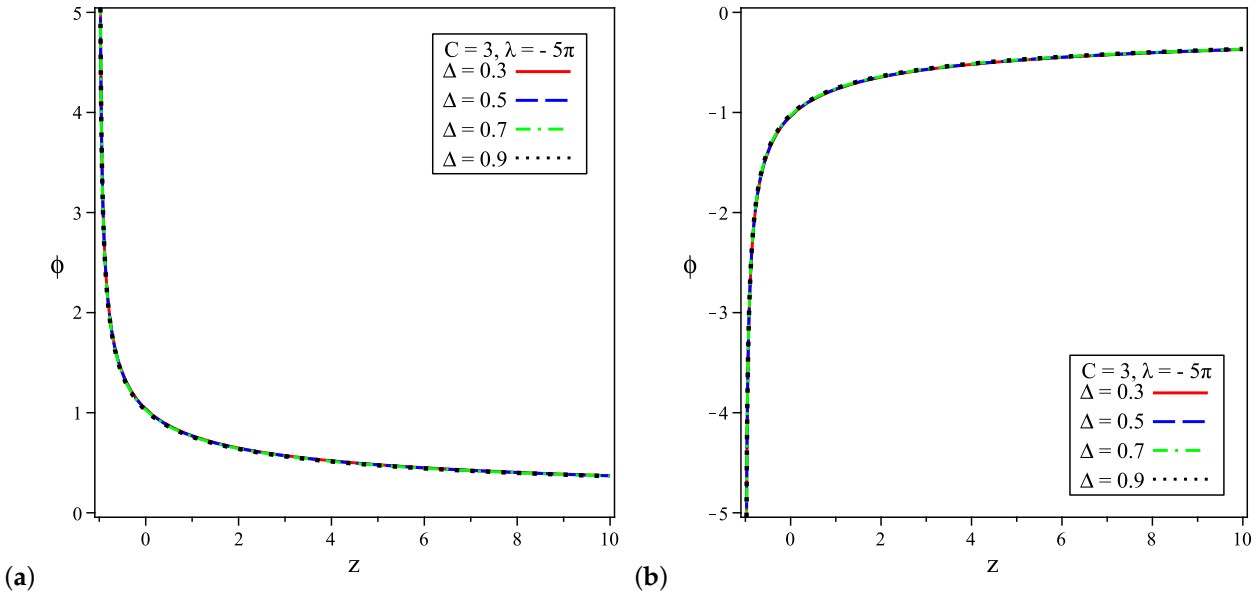

**Figure 4.** Depicting the nature between the redshift ($z$) and the Barrow holographic $k$ essence: (**a**) ($\phi$) is positive and (**b**) ($\phi$) is negative; here, $C = 3$ and $\lambda = -5\pi$.

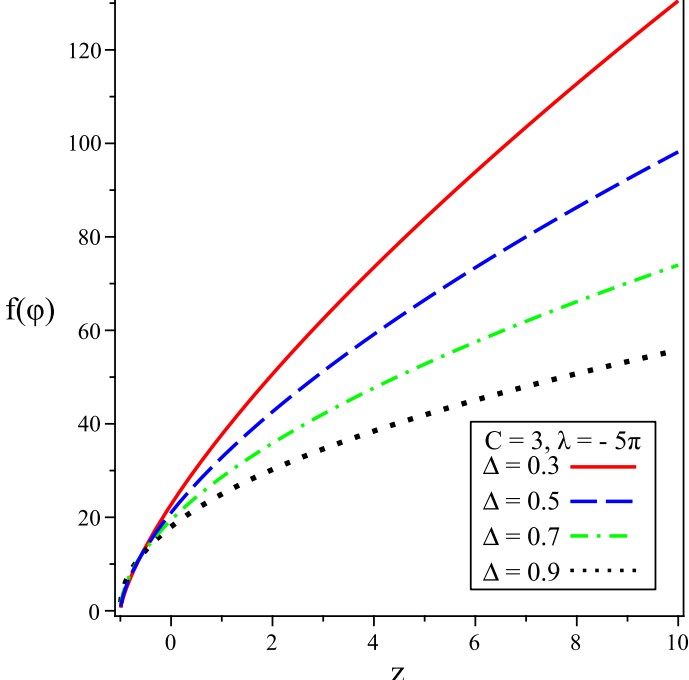

**Figure 5.** The behaviour of the Barrow holographic $k$-essence potential against redshifts ($z$); here, $C = 3$ and $\lambda = -5\pi$.

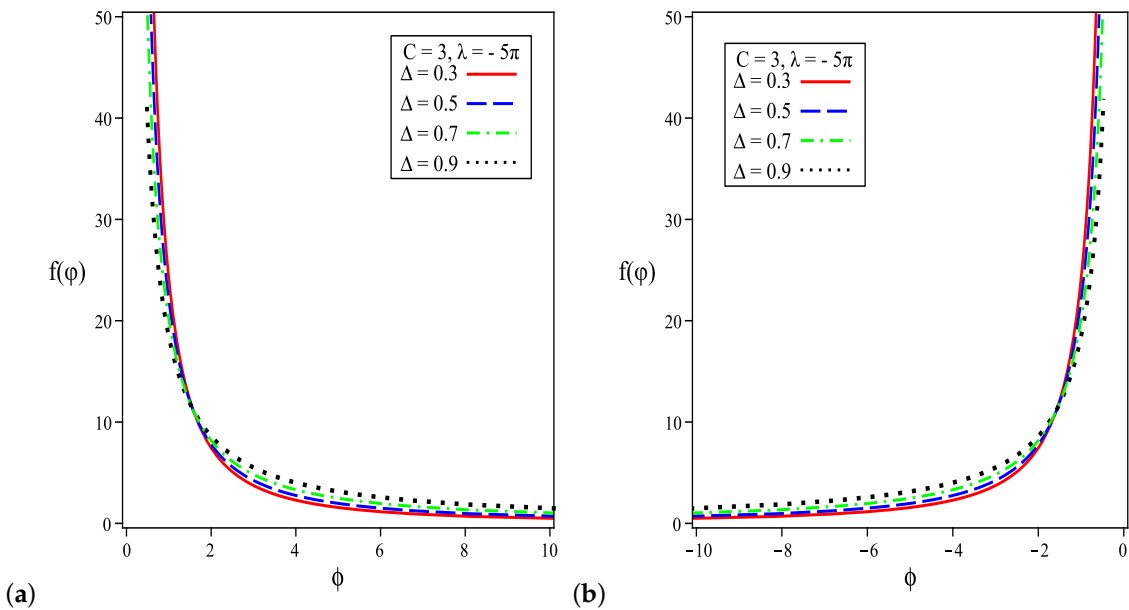

**Figure 6.** The plot of the Barrow holographic *k*-essence potential ($f(\phi)$) against the scalar field ($\phi$): (**a**) ($\phi$) is positive and (**b**) ($\phi$) is negative. Here, $C = 3$ and $\lambda = -5\pi$.

*5.3. Dilation Model for Barrow Holographic Dark Energy*

The dilation field is used to evade some quantum instabilities concerning the phantom DE field and to describe the DE problem [108]. The Lagrangian density of the dilatonic DE corresponds to the pressure density of the scalar field, and it has the following form [103]:

$$p_d = -X + ce^{\theta\phi_d}X^2, \tag{28}$$

where $c$ and $\theta$ are the positive constants, and $X = \frac{1}{2}\dot{\phi}_d^{\,2}$. This model appears from a four-dimensional effective low-energy string action and includes higher-order kinetic corrections to the tree-level action in low energy effective string theory [103]. The correspondence between the energy density of BHDE given by Equation (3) and the dilation energy density $\rho_d = -X + 3ce^{\theta\phi_d}X^2$ [103] can be seen as

$$\rho_d = -X + 3ce^{\theta\phi_d}X^2 = CH^{2-\Delta}, \tag{29}$$

and the correspondence of the Barrow holographic dark energy EoS is seen as

$$\omega_d = \frac{-1 + ce^{\theta\phi_d}X}{-1 + 3ce^{\theta\phi_d}X} = -1 - \frac{(\Delta - 2)}{2}\left[\frac{8\pi + 2\lambda}{8\pi + 3\lambda}\right]. \tag{30}$$

Concerning $\phi_d$, solving this equation and integrating $t$, we receive the following.

$$\phi_d = \frac{2}{\theta}\log\left[\frac{\theta}{\sqrt{2c}}\sqrt{\frac{4\lambda + 4\pi\Delta + \lambda\Delta + 8\pi}{8\pi + 6\lambda + 12\Delta\pi + 3\Delta\lambda}}\,t\right], \tag{31}$$

In [103], we can see the existence of the estimating solution for the dilation and marked that the solution resembles $Xe^{\theta\phi_d}$ = constant for this case, which exhibits solution $\phi_d(t) \propto \ln t$. We obtained the results by the dilation field, and the comparison with the EoS parameter of BHDE is consistent with the results found in [103]. We plotted the graph in Figure 7 for the dilation scalar field, where Figure 7a represents the positive scalar field and Figure 7b represents the negative scalar field. Fundamentally, we can witness the similarity in comparing the appeared dilation field with the point of *k*-essence, as shown in Figure 3. The version of the dilation field corresponds to [104]. We can depict $\theta^{2-\Delta}(e^{\frac{\Delta\theta\phi_d}{2}})$ in terms of

$\Delta$ and $\lambda$ upon substituting $Xe^{\theta\phi_d}$ from Equation (30) and operating Equations (11) and (31) in Equation (29).

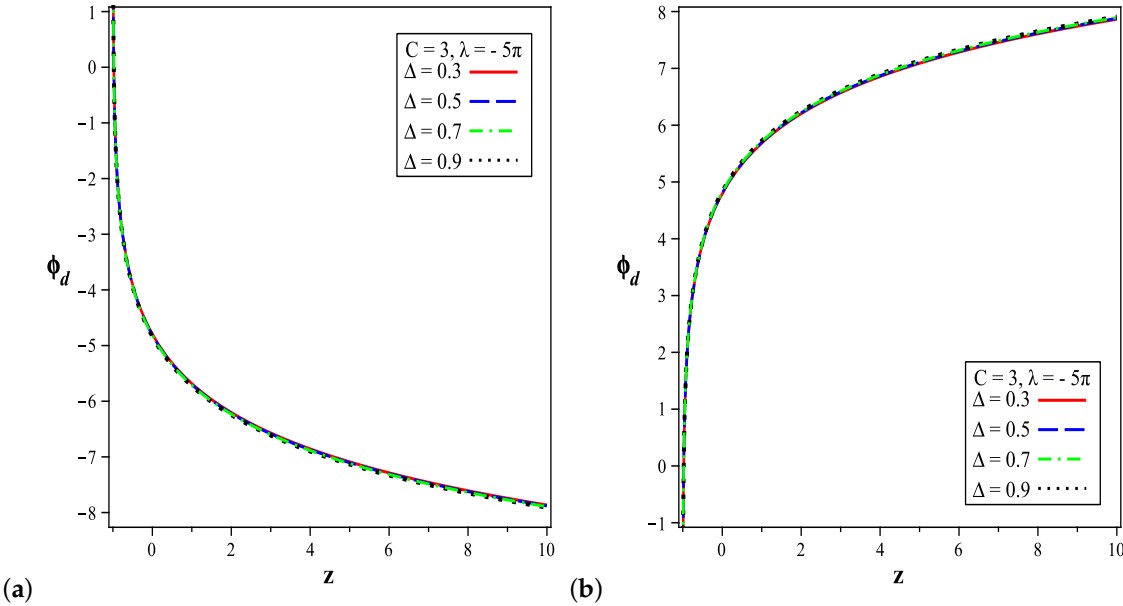

**Figure 7.** Depicting the nature between the redshift ($z$) and the scalar field of the Barrow holographic dilation ($\phi_d$), (**a**) ($\phi_d$) is positive and (**b**) ($\phi_d$) is negative; here, $C = 3$ and $\lambda = -5\pi$.

$$\theta^{2-\Delta}(e^{\frac{\Delta\theta\phi_d}{2}}) = \frac{C\ 2^{\frac{\Delta}{2}}}{c^{\frac{\Delta-4}{2}}}\left[\frac{8\pi+3\lambda}{8\pi+6\lambda+12\Delta\pi+3\Delta\lambda}\right]$$

$$\left[\frac{4\lambda+4\Delta\pi+\Delta\lambda+8\pi}{8\pi+6\lambda+12\Delta\pi+3\Delta\lambda}\right]^{\frac{\Delta}{2}}\left[\frac{2}{3}\left(\frac{8\pi+3\lambda}{8\pi+2\lambda}\right)\right]^{\Delta-2}. \tag{32}$$

The condition for an accelerated expansion of the universe provides $1/2 < cXe^{\theta\phi_d} < 2/3$, and for $\Delta = 2$ and $\lambda = -4\pi$, it corresponds to the cosmological constant, which is $\omega_\Lambda = \omega_d = -1$.

## 6. Conclusions

In this work, we explored the evolution of the universe in the framework of $f(R,T)$ gravity for the Barrow holographic quintessence, $k$-essence, and dilation models using the Hubble horizon as the IR cut-off. This investigation shows three major parameters: $\Delta$, $C$, and $\lambda$. For the accelerated expansion of the universe, parameter $\Delta$ of BHDE plays a significant role. For $-6\pi < \lambda < -4\pi$ and $0 < \Delta < 1$, we see that the EoS of BHDE explores the quintessence, $k$-essence, and dilation field with $\omega_\Lambda > -1$, and for $-6\pi < \lambda < -4\pi$ and $\Delta > 2$, it presents phantom phase $\omega_\Lambda < -1$, whereas the EoS of BHDE imitates cosmological constant $\omega_\Lambda = -1$, when $\lambda = -4\pi$ or $\Delta = 2$. For the BHDE model in phase $\omega_\Lambda > -1$, it is a general region for the $k$-essence, quintessence, and dilation era. To demonstrate the correspondence with $k$-essence, dilation and quintessence field, we constructed the scalar field and potential field.

In this study, we make graphs of the scalar field and the potential, which are displayed as Figures 1–3 for quintessence, Figures 4–6 for the $k$-essence field, and the dilation field is shown in Figure 7. We plotted the graph of the scalar field versus redshifts and the potential versus the scalar field, taking both the positive and negative signs of the scalar field for the quintessence (Figures 1 and 3), $k$-essence (Figures 3 and 6), and dilation (Figure 7) models. Figures 2 and 3 show the plot of the potential versus redshift for quintessence and $k$-essence models, respectively. It is clear from Figure 1a that the scalar field $\phi_q$ decreases as redshift $z$ increases. It remains constant with the

increasing redshift, evolving to a finite value in the future. A similar nature was observed for scalar field $\phi_q$ in [104] for the *k*-essence scalar field [102]. For the negative scalar field, Figure 1b explains the shift in the scalar field. Due to this result, the shape of the potential can be changed without having any effect on the cosmological constant. Figure 2 shows that the quintessence potential field diminishes if we shift from past to future. The same nature was marked for the potential of the model of holographic quintessence in [77] and the model of tachyon in [76].

Figure 3a explains the scalar field as the potential reduces when the scalar field is positive. The authors [105] showed the same nature for the quintessence potential. In contrast, when the scalar field is negative, Figure 3b shows that in the past: the potential is steeper. In the future, it has a flat approach, which means that the quintessence field proceeds with the decreasing potential with the expansion of the universe. [77] gave a similar result; hence, [105] suggested the potential drops with the evolution of the universe and that they are a runway type. Figures 3a and 7a express increments in redshifts, *z*, as the scalar field decreases, and at low redshifts, it becomes finite, which is discussed in [104] with the GO cut-off for the holographic *k*-essence. In contrast, Figures 3b and 7b present an increase in the scalar field with a decrease in the redshift. The version also corresponds to [104]. Figure 5 portrays the decrease in the potential if we roll from the past to the future. From Figure 6a, we can note that the potential drops with an expansion in the scalar field, whereas we can see the increment in the potential with a decrease in the scalar field in Figure 6b. A similar nature with respect to the potential in [104] can be observed in Figures 5 and 6.

The reconstruction results in this research study are promising for describing the main characteristics of the potential field and the scalar field for the quintessence, *k*-ssence, and dilation models. Even though BHDE promotes the reconstruction unambiguously and quickly, one must explore and understand the theoretical source of Barrow holographic densities. This research study resolved the reconstruction of models at the phenomenological level, and the theoretical roots of Barrow holographic densities need to be examined.

**Author Contributions:** Writing—original draft preparation, G.V.; Formal analysis, M.K.; Writing—review and editing, U.K.S. All authors have read and agreed to the published version of the manuscript.

**Funding:** This research received no external funding.

**Institutional Review Board Statement:** Not applicable.

**Informed Consent Statement:** Not applicable

**Data Availability Statement:** The present work is theoretical study, and therefore there is no data will be deposited.

**Acknowledgments:** The author U. K. Sharma is thankful towards the IUCAA, Pune, India, for awarding a visiting associateship. We sincerely thanks the anonymous reviewer for his constructive comments that enhanced the quality of the paper.

**Conflicts of Interest:** The authors declare no conflict of interest.

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
