# Peer review of "Scalar Field Models of Barrow Holographic Dark Energy in f(R,T) Gravity"

_universe, doi:10.3390/universe8120642_

Round 1

Reviewer 1 Report

The manuscript is poorly presented. However, there is no physical motivation for this modified gravity model. Several typos and grammar errors (even in the section titles) exist. However, I do not see the point in mixing Barrow entropic model with f(R, T), scalar fields, k-essence, etc. Barrow entropic model aims to find alternative dark energy models without considering Dark Energy perse. The motivation and interest of this research are borderline and do not add any new results to the vast subject of dark energy models. I suggest rejection. 

Author Response

We would like to sincerely appreciate your evaluation on our manuscript. We have rewritten the manuscript for improving the presentation. We have increased the motivation and interest of this research which can be seen on page-3 in bold as In this work................................................problems of current interest.”

Reviewer 2 Report

Referee Report

The authors study the holographic dark energy motivated by the Barrow entropy and its correspondence with various scalar field models. The comments are given below.

1) It is very difficult to understand why the authors have taken the f(R, T) gravity in the background. If the motive is to study the correspondence with scalar field models, what is the necessity of considering the f(R, T) gravity? Why is the EoS parameter stated through the f(R,T) gravity? Do we need it? It is very confusing to understand the role of f(R,T) here.

Moreover, Barrow holographic dark energy should itself be sufficient to drive the late cosmic acceleration. Why do we need f(R,T) gravity for that? Are the authors not sure about the capabilities of the Barrow holographic dark energy model? f(R,T) gravity involves the non-minimal coupling of the curvature and matter sector. Due to this, an extra force appears, which makes the motion of the particles non-geodesic in nature. This will basically imply the violation of the Equivalence principle. That is why f(R, T) gravity has been considered ineffective by many scientists. Whenever one works with this theory, one must address these issues from the point of view of their work. The authors completely ignore this. In the absence of this analysis, any work with f(R, T) gravity is incomplete.

2) I do not understand the statement before equation (6). What do the authors mean by “By changing the work of g_{\mu\nu} which is shown in Eq. (4)….”. We get field equations by varying the action with respect to the metric. How do we get it by changing the work of the metric? What work are they talking about? From the previous statement in the same paragraph probably the authors want to say that they consider f(R, T)=f(R)+2f(T). Why do the authors consider this model out of so many models discussed in Harko et al. 2011 [ref. 62]? This model does not even produce the non-minimal coupling between R and T.

3) After equation (6) the authors consider T=\rho whimsically. How do they consider this? We know that T=g^{\mu\nu}T_{\mu\nu}. We need to calculate T from eqn.(5) and the metric, which the authors have not done.

4) The authors take the pressure of matter as 0. This must be normal matter since it is pressure-less. It cannot be the dark energy as it has the negative pressure driving the acceleration. Then how do they suddenly land up with \rho_{\Lambda} in eqn.(8)? What is the relation between \rho and \rho_{\Lambda}? Moreover, what is \rho_{\Lambda} itself? Nothing has been stated anywhere.

5) After eqn.(5) they write \Lambda_{eff}. What is that?

6) The paper lacks a clear motivation. What is attempted and what is done do not quite match with each other. Moreover, why is the work done at all? When we have the literature flooded with works of BHDE in different modified gravity theories like Horava-Lifshitz gravity, and Brans-Dicke gravity, why again do we need another such work in a different gravity? What new do we get?

7) The paper is very poorly, carelessly, and whimsically written. In the heading of section 2, the spelling of Barrow is incorrect. Section 2 does not start with a capital letter. Throughout the paper at many places the word ‘Barrow’ have been written with a small letter ‘b’. It is a proper noun, so it should start with a capital letter. After every equation ‘here’ is written with a small letter ‘h’. It does not look good. After eqn.(9) Hubble has been written with a small ‘h’. In the acknowledgement, they write “The author is thankful …..”. There are three authors. Which author? There are numerous typos. Just before eqn.(15), the authors write “For thr quintessence field…”. The previous line starts with a “we” in the small letter “w”. In equation (15) there is a dot before \rho_{q}. There are numerous such problems in the paper. The authors did not check the paper carefully before submitting it.

 Before even going into the details of the work I found so many issues with the paper. This paper does not qualify to be a research article of any sort in its current form and needs to be rewritten. So it cannot be recommended for publication.

Author Response

We would like to sincerely appreciate your evaluation on our manuscript. Please find our response to reviewer's comments in the attached document.

Reviewer 3 Report

 Barrow holographic dark energy has attracted growing interest in the last decades. What was clear soon, is that can find the accelerated expansion in the frame of f (R, T) gravity. In this paper the
authors study scalar field models of Barrow holographic dark energy in f (R, T) gravity

The paper extends numerous papers. The subject is interesting, and the results are reasonable. There are, however, some weaknesses related to the presentation, which is rather poor. Here are a few questions and some recommendations for the authors.

- The equations are not well written, for example, Eq.1, Eq,4, Eq. 7...
their forms can be written in a more elegant manner.
- The reference of the equations must be adjusted, for example, Eq.ref eq1, and after Eq23....

The main remark and question are about the impact of the barrow parameter \Delta on the plot of the scalar field \phi vs the redshift z in Figures 4 and 7. The curves seem to be confounded, which suggests that \Delta has no effect. Maybe realizing a zoom can disclose such an issue or more discussion is needed.

Summarizing: the subject of the present paper could be of interest for
Universe, and also the results seem interesting, but the presentation is really awful. The paper should be carefully rewritten. I encourage the authors to improve heavily the quality of the presentation: check
English language, and typos, and also improve the presentation of the
formulas. Explanations should be more clear, the figures 4and 7
could be substantially improved and better integrated with the rest of
the paper.

I recommend a careful rewriting of the paper. After that, I can
reconsider my recommendation for publication in Universe. Please, do not provide "surgical changes" to the paper but, if possible, do all the efforts to improve the quality of the paper. To publish a well-presented paper is in the interest of both the parts, the journal and
the authors.

Author Response

We would like to sincerely appreciate your kind evaluation on our manuscript. Please find our response to reviewer's comments in the attached document.

Round 2

Reviewer 1 Report

The manuscript has been substantially improved. The authors can cite Barrow Entropy Cosmology: an observational approach with a hint of stability analysis, 

e-Print: 2108.10998 [astro-ph.CO], DOI: 10.1088/1475-7516/2021/12/032

Published in: JCAP 12 (2021) 12, 032.

Other relevant works about Barrow entropy can be cited. 

Reviewer 2 Report

The response given by the authors is totally inappropriate. They do not make any sense, nor do they have any connection with the questions that I raised. For example, the first question that I raised is not answered at all. The authors did nothing about the non-geodesic motion of the test particles. They did not provide a concrete motivation for choosing the f(R,T) gravity.  They simply added some statements in various places in the paper. No equations were changed and nothing concrete was performed following the comments. The same holds for the other questions. The authors simply added some texts which are irrelevant to the comments raised. 

Moreover, I do not know why this manuscript was sent to me for a repeat review. I did not give any revision option. It was a straight rejection, which holds now as well. I do not recommend the paper for publication based on the reply given by the authors.

Reviewer 3 Report

I recommend this version for publication